# Optimization of Size of Nanosensitizers for Antitumor Radiotherapy Using Mathematical Modeling

**DOI:** 10.3390/ijms241411806

**Published:** 2023-07-22

**Authors:** Maxim Kuznetsov, Andrey Kolobov

**Affiliations:** P.N. Lebedev Physical Institute of the Russian Academy of Sciences, 53, Leninskiy Prospekt, Moscow 119991, Russia; kuznetsovmb@mail.ru

**Keywords:** mathematical oncology, numerical optimization, radiosensitizers, nanoparticles

## Abstract

The efficacy of antitumor radiotherapy can be enhanced by utilizing nonradioactive nanoparticles that emit secondary radiation when activated by a primary beam. They consist of small volumes of a radiosensitizing substance embedded within a polymer layer, which is coated with tumor-specific antibodies. The efficiency of nanosensitizers relies on their successful delivery to the tumor, which depends on their size. Increasing their size leads to a higher concentration of active substance; however, it hinders the penetration of nanosensitizers through tumor capillaries, slows down their movement through the tissue, and accelerates their clearance. In this study, we present a mathematical model of tumor growth and radiotherapy with the use of intravenously administered tumor-specific nanosensitizers. Our findings indicate that their optimal size for achieving maximum tumor radiosensitization following a single injection of their fixed total volume depends on the permeability of the tumor capillaries. Considering physiologically plausible spectra of capillary pore radii, with a nanoparticle polymer layer width of 7 nm, the optimal radius of nanoparticles falls within the range of 13–17 nm. The upper value is attained when considering an extreme spectrum of capillary pores.

## 1. Introduction

Improving the efficacy of antitumor therapy requires not only the introduction of new technologies into clinical practice, but also the rationalization of the use of implemented agents and techniques. In practice, solving the problem of optimizing any type of therapy formally requires an enormous number of clinical trials, which is physically unfeasible and associated with insurmountable ethical difficulties, since the result of changing a clinical protocol may well reduce the overall treatment efficacy. Mathematical modeling, in which a tumor and its microenvironment are represented as a single dynamic system, can be of great help in solving complex optimization problems of this kind. This approach can narrow down the range of potentially effective treatment regimens that need to be tested in practice. Mathematical oncology is currently gaining popularity and has already led to some positive results in both preclinical and clinical settings [1]. Ideally, the corresponding optimization tasks have to imply the enormous complexity and dramatic variability of cancers, their constant evolution, and the impossibility of performing reliable estimations of all the related parameters during an ongoing treatment. That drastically distinguishes optimization tasks in mathematical oncology from the optimization tasks in exact sciences and computer science [2,3].

The two most common therapeutic methods in oncology are surgery and radiotherapy. Given the widespread use of radiotherapy and its applicability to the vast majority of tumors, radiobiologists have suggested that optimizing radiotherapy is a more effective way than developing expensive tumor-specific drugs to achieve a comparable increase in the worldwide cure rate for cancer [4]. One promising technology in this context is the use of nonradioactive nanosensitizers that emit secondary radiation when activated by the primary beam. When accumulated in the tumor, nanosensitizers allow localizing and increasing the effective dose of radiation directly in the malignant tissue. The overall efficacy of the entire course of radiotherapy with nanosensitizers should depend significantly on a number of factors, including the schedule of their administration and the distribution of irradiation both in time (fractionation) and space (dose painting). Moreover, the crucial factor determining the success of treatment is the efficiency of the delivery of the active radiosensitizing agent to the tumor. In order to increase the therapy efficacy and minimize its side effects, a rational approach is to use tumor-specific nanoparticles, each of which represents a small volume of the active substance covered by a layer of polymers with specific antibodies embedded in it.

However, the use of such nanoparticles is associated with certain difficulties in their delivery to the tumor when injected intravenously into the body. The inflow of large particles into the tumor, in particular, is impeded by the necessity of their penetration through the small pores in the capillary walls (this factor is somewhat facilitated but not eliminated by the proangiogenic substances produced by the tumor) [5], the localization of newly formed tumor capillaries, which are mostly not at the depth of the tumor, but at its interface with normal tissue [6], and the increased interstitial fluid pressure in the tumor, which limits the advective inflow of nanoparticles from the capillaries and leads to washout of fluid from the tumor into surrounding tissue [7]. Defining the appropriate size for the production of nanosensitizers can be facilitated by mathematical modeling.

In order to create the model that provides the quantitative answer for the optimal size of nanoparticles, a reasonable approach is to account for all the known physiological processes that can eventually influence the distribution of nanoparticles within the tumor. Crucially, the process and peculiarities of tumor angiogenesis must be considered in a detailed and physiologically sound manner. Interstitial fluid dynamics has to be accounted for, as fluid transports nanoparticles from blood to tumor, redistributes them through the tissue, and carries them away from the tissue into the lymphatic system. The correct method for modeling the dynamics of interstitial fluid is to simultaneously consider the stress arising in the solid phase of the tissue, as it is closely related to fluid pressure—in particular, the deformation of the solid component of the tissue affects fluid flow, while fluid outflow from the tumor leads to its shrinking and solid stress relaxation.

At present, mathematical modeling of tumor growth taking into account the biomechanical aspects is an unpopular area of research, in particular, because of the relatively recent recognition of the significance of their influence on tumor growth. The first relevant experimental study was conducted in 1997 [8], and the first corresponding works on mathematical modeling appeared in 2003 [9,10]. In these, tumor tissue is treated as a fluid-like medium or as an anisotropic linear elastic medium, but the interstitial fluid is not taken into account. Despite the relative simplicity of such approaches, their use makes it possible to reproduce key experimental observations, in particular, the effect of maximum tumor size reduction with an increase in external pressure. More sophisticated approaches, adapted from continuum mechanics and based on the multiplicative decomposition of the tissue gradient deformation tensor have now emerged [11,12]. The components of decomposition correspond to tumor proliferation described as the stretching of tumor tissue, to the elastic response of the tumor and normal tissue, and, in some works, to the formation of residual stress in the tumor [13] and to the reorganization of intercellular links in response to deformation [14,15]. However, such approaches are characterized by significant computational complexity and, as a consequence, are inapplicable in practice for solving optimization problems that require a large number of simulations under variation of multiple model parameters. Moreover, apparently due to their computational complexity, as far as we know, mathematical models of tumor growth taking into account biomechanical aspects have not yet been used for the explicit modeling of antitumor therapy as regards reproducing the reaction of the tumor and its microenvironment to therapeutic action. This is particularly relevant to the capillary network. In addition, there exist the works devoted to modeling the distribution of a therapeutic agent in the tumor after its injection into the blood, which is a much less computationally expensive task [16,17].

Based on our models that were previously used to study various aspects of tumor growth and treatment [18,19,20] and based on the elements of ideological approaches used in the above articles, we developed a mathematical model of tumor growth and its radiotherapy with the use of intravenously administered nanosensitizers. We solved the problem of finding the optimal size of nanoparticles to achieve maximum radiosensitization of the tumor as the result of a single injection of a given total volume of particles. The resulting model is based on the use of essential simplifications applicable to the function of the deformation of a set of interconnected cells when considering a spherically symmetric problem. These simplifications should make it practically possible to use the developed model to further solve the problems of the spatiotemporal optimization of antitumor radiotherapy using radiosensitizing nanoparticles, and as a basis for solving other similar problems.

The computational code of the model was implemented in C++ and can be found in the Appendix A.

## 2. Results

### 2.1. Mathematical Model

The developed mathematical model is presented in detail in Section 4. Here, the main information is outlined.

The block scheme of the main model interactions is presented in Figure 1. It depicts 11 spatially distributed variables of the model. Three of them correspond to different types of tumor cells: proliferating, quiescent and damaged. Normal cells and interstitial fluid are also present in the tissue section under consideration. Two types of capillaries are considered: normal and abnormal. The latter have altered properties due to the influence of vascular endothelial growth factor, VEGF. Glucose was chosen as the key nutrient since it is the essential element for biosynthesis and the key energy metabolite for tumor cells. The nanoparticles in the tissue exist in both a free state and a tumor cell-bound state. There is also one time-dependent variable in the model, which determines the concentration of nanoparticles in the blood. Its instantaneous increase corresponds to the injection of particles into the blood. Another external action corresponds to instantaneous irradiation, resulting in the transition of tumor cells to a damaged state.

The model considers the spherically symmetric growth of a noninvasive tumor in normal tissue. The tumor and normal tissue consist of two phases: the cells, forming a solid porous fraction, and the intercellular fluid, capable of flowing through the pores of the solid fraction. The tissue is assumed to be saturated and incompressible, that is, the total density of cells and fluid is constant and, for convenience, is normalized to unity. Cell proliferation happens with the use of intercellular fluid (and the implicitly implied substances dissolved in it) as a source of mass, and cell death is reflected by the reverse transition. The rate of cell proliferation depends on the rate at which they consume glucose, and on the local solid stress. When glucose levels drop significantly, tumor cells switch to a quiescent state, and this transition is reversible. Cell death due to the lack of nutrients for simplicity is not considered in this paper. Fluid flows from the capillaries by filtration through their pores and drains into the lymphatic system, which is not explicitly considered, but it is assumed that the density of the lymphatic capillaries is proportional to that of normal cells. The intercellular fluid flows through the porous solid phase according to Darcy’s law, assuming that all external forces (e.g., gravity) can be neglected, that the flow of fluid through the pores is slow enough so that inertial effects become negligible, and that the solid phase is uniformly permeable. For simplicity, it is assumed that the solid stress is isotropic, that is, it acts with equal magnitude in all directions. Thus, the solid phase behaves as an elastic fluid substance. It then follows from the theory of porous media that the pressure gradients of liquid and solid stress are equal to each other with opposite sign [21].

The solid stress function, illustrated in Figure 2, is based on a rather general assumption that the volume fraction of cells is related to the average distance between them [22]. It is a smooth function that is qualitatively consistent with experimental observations [23]. It is assumed that when the fraction of cells $s=s_{0}$ is normal, cell interactions result in zero solid stress. As the cells become closer together, $s>s_{0}$, there is a repulsive interaction, σ>0. As the distance between the cells increases, $s<s_{0}$, an attractive interaction emerges. At first, it intensifies and then it gradually weakens due to successive ruptures of individual intercellular contacts. The stress becomes zero at $s=s_{s}<s_{0}$.

The method of considering tumor angiogenesis is based on the approach that we introduced in the work [18]. VEGF is produced by quiescent cells under metabolic stress, it diffuses through the tissue, nonspecifically degrades, and binds to endothelial cells, affecting capillary properties and triggering the process of angiogenesis. The model uses two variables to describe the capillary network, which are introduced separately to account for differences in permeability of preexisting capillaries and capillaries affected by VEGF. These variables have a physical meaning of capillary surface density, which is used to facilitate the description of the transvascular transport of substances. It is assumed that capillaries lose their functionality and degrade within the tumor due to their rupture, which is caused by their displacement and chemical factors [24]. The action of VEGF on capillaries leads to angiogenesis, which is described by the formation of abnormal capillaries based on both types of existing capillaries, and to the “denormalization” of capillaries, which is introduced into the model to reflect the increase in capillary permeability under the action of VEGF. Both actions are modeled by the terms of the Michaelis–Menten type, as a result of which the rates of the corresponding processes vanish in the absence of VEGF and approach maximum values when it is abundant. At low VEGF concentrations, the capillaries are normalized and the opposite transition occurs. The overall density of the capillary network is limited from above. In reality, while capillaries are formed during angiogenesis, they sprout in the direction of increasing VEGF concentration, and in the model, this is reflected by the active movement of abnormal capillaries. In addition, the capillaries move together with the advective flow of the solid phase of the tissue.

Glucose flows from the capillaries into the tissue, diffuses through it, and is consumed by the cells, with proliferating tumor cells consuming it much faster than others. Since the vast majority of glucose enters the tissue by diffusion through the capillary walls [5], only this type of transvascular transport is considered.

The model considers spherical radiosensitizing nanoparticles of radius ξ, covered by a polymer layer of width ψ<ξ. It is assumed that under this layer, there is active substance, which affects the efficiency of irradiation when the particles bind to tumor cells. For simplicity, it is assumed that the local concentration of radiosensitizing agent in the particles bound to cells leads to a linear increase in the radiation dose. The damage to cells as a result of radiotherapy is described by a standard linear quadratic model [4].

The particle size ξ is varied in this work to determine their optimal size for the maximum radiosensitization of the tumor. The same total volume of nanoparticles is injected into the blood; hence, the larger the particles, the larger the total fraction of the active substance in them. The size of the particles also affects their dynamics, as determined by Equation (3).

Firstly, large particles have a lower diffusion coefficient, which affects both their movement through tissue and the diffusion component of their inflow from capillaries. Secondly, nanoparticles are eliminated from the blood with a characteristic time in the order of hours (due to the filtration by the liver), with large particles being eliminated from the body faster in accordance with experimental observations [25]. The coefficient of binding to tumor cells is considered to be independent of the particle size. Thirdly, nanoparticle size influences their diffusive permeability and the fraction of the pore cross-sectional area available for them in the walls of capillaries of different types, as Figure 3 illustrates.

The resulting capillary permeability for nanoparticles is determined by the convolution of certain functions on the spectra of pores in capillaries walls. For advective inflow, it is a function related to the steric exclusion, which characterizes the fact that only a certain fraction of the pore cross-sectional area is available for particle movement. The function of diffusive inflow takes this into account, and also the diffusion coefficient of nanoparticles and the hydrodynamic resistance that they encounter as they move through the pores, which is approximated by the experimentally derived Renkin equation [5]. The total number of pores in capillaries of different types is matched to the experimental data on capillary permeability for glucose in the absence of VEGF and under its influence. From these estimations, it reasonably follows that the larger the particles, the worse they pass through any capillaries, but angiogenic capillaries are less restrictive in limiting their inflow.

### 2.2. Free Growth and Irradiation of Tumor

At the very beginning of the model simulation, all tumor cells proliferate while they actively consume glucose, and within a few days, its deficit is established inside the tumor, stimulating the transition of cells to a quiescent state. In this state, they begin to produce the proangiogenic factor VEGF. It diffuses through the tissue, binds with capillaries, and leads to two key effects. First, the existing capillaries transit from a normal state to an abnormal state with increased permeability, and second, new abnormal capillaries are formed. The red dashed line in Figure 4 shows the sum of both types of capillaries. However, within the tumor, the capillaries are destroyed, so the lack of glucose in the tumor core persists, in contrast to its outer layer. This is qualitatively consistent with the experimental observations on the location of functional capillaries in sufficiently large tumors [6].

Increased capillary permeability also leads to enhanced fluid inflow into the tissue, but in normal tissue, the lymphatic system drains excess fluid effectively, so its level there remains almost unchanged. Inside the tumor, there are no lymphatic capillaries, so the fluid level in the tumor increases, as does its pressure. This is highlighted in the right figure where the lymphatic capillaries are absent. Thus, the fluid pressure inside the tumor is elevated, and it is normalized at the border with normal tissue. This also corresponds well with experimental data and has been previously reproduced in mathematical models [7].

The lower figure shows the distribution of the local fraction of cells that wold be affected by radiation if it is administered at the designated moment. Since nonproliferating tumor cells are less sensitive to radiation, a larger proportion of cells would survive deep within the tumor.

Figure 5 demonstrates the result of tumor irradiation. It leads to the formation of a population of damaged cells, which further die, from the point of view of the model turning into a fluid flowing out of the tumor against its own pressure gradient. As a result, the tumor slightly shrinks at first, but in the absence of further irradiation, it recovers within a few hours and continues to grow.

### 2.3. Injection of Nanoparticles

Figure 6 demonstrates the result of injection of nanoparticles into the blood at the same moment in the absence of irradiation. In the model, the introduction of nanoparticles is reflected by a sharp increase in their concentration in the blood, which then gradually decreases due to the particles clearance from the body. As long as this concentration is high, the terms of nanoparticle inflow into the tissue are acting actively. The particles move through the tissue by means of both diffusion and advection, the latter, in particular, causing them to be partially washed out from the tumor into the normal tissue. Some of the particles are bound to tumor cells, becoming immobilized, i.e., they stop moving relative to them. The presence of bound particles increases the radiosensitivity of the cells, and therefore, the tumor radiosensitivity grows during the first hours after the injection of nanoparticles. However, soon, the terms of the outflow of free nanoparticles from the tissue, through the blood and the lymphatic system, also become efficient. After a time of about two or three days, the tumor boundary is occupied by the newborn cells to which the particles have not had time to bind, so the radiosensitivity of the outermost cells starts decreasing. In addition, the cells to which the nanoparticles have bound become quiescent as they move away from the tumor border due to the drop in the local glucose level, and their radiosensitivity also decreases as a consequence of it.

### 2.4. Optimization of Nanoparticle Size

Figure 7 demonstrates the dynamics of the total amount of bound active substance and the fraction of potentially damaged tumor cells as the size of injected nanoparticles varies (they are also injected on day 51 of tumor growth), with a pore spectrum of abnormal capillaries $PS_{a}\left(x\right)$, defined in Equation (3). According to the description of tumor dynamics and the nanoparticle distribution given above, when enough nanoparticles penetrate the tumor, its radiosensitivity at first increases, and then gradually decreases. However, as follows from the graphs, at the particle sizes of 30 nm and higher, the increase in radiotherapy efficacy in this case is negligible. In general, 30-nanometer particles are able to bind to the tumor when administered in a two orders of magnitude lower amount of active substance than 10- and 15-nanometer particles.

The results suggest that for the maximal radiosensitization of the tumor, the particles should be relatively small, but not too small, otherwise they will contain too little of the active substance. Given the selected abnormal capillary pore spectrum and other selected model parameters, the maximum radiosensitization is achieved for a nanoparticle radius of 13 nm, with its variation by two nanometers resulting in the reduction in the maximum fraction of potentially damaged cells by no more than three percent.

The pore spectrum of abnormal capillaries is obviously a factor that significantly influences this result, and the model calculations allow us to investigate to what extent an increase in capillary permeability can shift the optimal nanoparticle size. Figure 8 shows similar results for more permeable abnormal capillaries, with a pore spectrum $PS_{b}\left(x\right)$. In this case, 50-nanometer particles allow as much active agent to accumulate inside the tumor as 30-nanometer particles do for the previous spectrum $PS_{a}\left(x\right)$. But such large particles still do not result in any notable increase in irradiation efficacy. The important conclusion is that, while the use of large, e.g., 30-nanometer particles, becomes noticeably more effective when the pore spectrum is changed, the efficiency also increases for smaller particles. As a result, the optimal particle radius increases, but not significantly, to 15 nm.

It is not clear from the literature data to what extent the pore spectrum can be shifted in a physiologically reasonable way. Figure 9 demonstrates the simulation results in the very extreme case of a uniform pore spectrum of abnormal capillaries $PS_{c}\left(x\right)=1$. In this setting, a pore of 100 nm radius is met in the abnormal capillary wall as often as a pore of 1 nm radius, and as a pore with any other radius between these values. Even in this case, the optimum particle radius does not shift much and becomes equal to 17 nm. This suggests that, whatever physiologically reasonable properties of abnormal capillaries are selected in the model, the optimal particle radius of 17 nm is a limiting value, at least if the other values of model parameters are maintained. Moreover, the graph of maximum irradiation efficacy with respect to varying particle size becomes more flat, i.e., a deviation of 2 nm already has a weaker effect on the optimal result, which is explained by the approaching complete eradication of tumor cells at the tumor border.

It is worth noting that, if the tumor stimulates angiogenesis weakly, the optimal particle size is, of course, reduced. Simulations under the complete absence of angiogenesis, not shown here, lead to the formally optimal nanoparticle radius of 10 nm, but then, in any case, so few particles penetrate into the tumor that the increase in radiosensitivity due to their action is negligible.

## 3. Discussion

The results of mathematical modeling show that the optimal size of nanoparticles for the maximum radiosensitization of the tumor during a single intravenous injection of a given total volume of nanoparticles depends on the degree of permeability of capillaries formed as a result of tumor angiogenesis and affected by proangiogenic factors produced by tumor cells. Given the physiologically reasonable values of the pore spectrum of such capillaries, the width of the polymer layer of nanoparticles surrounding the active substance of 7 nm, and physiologically reasonable other values of mathematical model parameters, the optimal radius of nanoparticles lies between 13 and 17 nm, with the upper value achieved using a very extreme pore spectrum of tumor capillaries, in which pores with any radii in the range of 1–100 nm are equally frequent. Reducing the size of nanoparticles negatively affects the efficacy of irradiation due to the decrease in the volume fraction of the active substance in them. Increasing the size of nanoparticles has a negative effect on the irradiation efficacy primarily due to the decrease in capillary permeability for them under any capillary pore spectrum, and to a lesser extent, due to the accelerated removal of nanoparticles from the body and due to the decrease in their diffusion coefficient, which complicates their penetration into the tumor from capillaries located predominantly at its border and outside.

The developed model will be used in the future to solve the problem of the spatiotemporal optimization of proton therapy using radiosensitizing nanoparticles. The low computational complexity of the implemented approach will allow hypotheses to be formed on therapy optimization, which is relevant in a wide range of tumor-specific and patient-specific parameters. The simplifications used during model development should not compromise the results, since the potential practical value lies primarily in the qualitative trends of changes in treatment settings that lead to their optimization.

## 4. Materials and Methods

### 4.1. Model Equations

The system of Equation (1) defines the dynamics of the model variables.

proliferating tumor cells: 
$\frac{\partial n_{p}}{\partial t}=\overset{\mathrm{proliferation}}{\overbrace{Bn_{p}\cdot \Theta_{p}\left(\sigma \right)\frac{g}{g+g^{*}}}}\overset{\mathrm{transition}}{\overbrace{-B\cdot \left[1-\Theta_{tr}\left(g\right)\right]n_{p}+B\cdot \Theta_{tr}\left(g\right)n_{q}}}\overset{\mathrm{irradiation}}{\overbrace{-R_{p}}}\overset{\mathrm{advection}}{\overbrace{-\frac{1}{r^{2}}\frac{\partial (I_{s}n_{p}r^{2})}{\partial r}}};$
quiescent tumor cells: 
$\frac{\partial n_{q}}{\partial t}=\overset{\mathrm{transition}}{\overbrace{B\cdot \left[1-\Theta_{tr}\left(g\right)\right]n_{p}-B\cdot \Theta_{tr}\left(g\right)n_{q}}}\overset{\mathrm{irradiation}}{\overbrace{-R_{q}}}\overset{\mathrm{advection}}{\overbrace{-\frac{1}{r^{2}}\frac{\partial (I_{s}n_{q}r^{2})}{\partial r}}};$

normal cells: 
$\frac{\partial h}{\partial t}=\overset{\mathrm{advection}}{\overbrace{-\frac{1}{r^{2}}\frac{\partial (I_{s}hr^{2})}{\partial r}}};$

damaged cells: 
$\frac{\partial m}{\partial t}=\overset{\mathrm{irradiation}}{\overbrace{R_{p}+R_{q}}}\overset{\mathrm{death}}{\overbrace{-Mm}}\overset{\mathrm{advection}}{\overbrace{-\frac{1}{r^{2}}\frac{\partial (I_{s}mr^{2})}{\partial r}}};$

interstitial fluid: 
$\frac{\partial f}{\partial t}=\overset{\mathrm{inflow}}{\overbrace{\left[L_{n}c_{n}+L_{a}c_{a}\right]\cdot \left[p_{c}-p\right]}}\overset{\mathrm{outflow}}{\overbrace{-L_{l}h\left[p-p_{l}\right]}}\overset{\mathrm{cell}\;\mathrm{death}}{\overbrace{+Mm}}\overset{\mathrm{cell}\;\mathrm{proliferation}}{\overbrace{-Bn_{p}\cdot \Theta_{p}\left(\sigma \right)\frac{g}{g+g^{*}}}}\overset{\mathrm{advection}}{\overbrace{-\frac{1}{r^{2}}\frac{\partial (I_{f}fr^{2})}{\partial r}}};$

VEGF: 
$\frac{\partial v}{\partial t}=\overset{\mathrm{secretion}}{\overbrace{S_{v}n_{q}}}\overset{\mathrm{internalization}}{\overbrace{-\omega [c_{n}+c_{a}]v}}\overset{\mathrm{degradation}}{\overbrace{-M_{v}v}}\overset{\mathrm{diffusion}}{\overbrace{+D_{v}\Delta v}}$

normal capillaries: 
$\frac{\partial c_{n}}{\partial t}=\overset{\mathrm{degradation}}{\overbrace{-M_{c}\left[n_{q}+m\right]c_{n}}}\overset{\mathrm{normalization}}{\overbrace{+\frac{V_{n}v^{*}}{v+v^{*}}c_{a}}}\overset{\mathrm{denormalization}}{\overbrace{-\frac{V_{d}v}{v+v^{*}}c_{n}}}\overset{\mathrm{advection}}{\overbrace{-\frac{1}{r^{2}}\frac{\partial (I_{s}c_{n}r^{2})}{\partial r}}};$

abnormal capillaries: 
$\frac{\partial c_{a}}{\partial t}=\overset{\mathrm{degradation}}{\overbrace{-M_{c}\left[n_{p}+k_{M}\left\{n_{q}+m\right\}\right]c_{a}}}\overset{\mathrm{angiogenesis}}{\overbrace{+\frac{Rv}{v+v^{*}}\left[c_{n}+c_{a}\right]\left[1-\frac{c_{n}+c_{a}}{c_{max}}\right]}}\overset{\mathrm{normalization}}{\overbrace{-\frac{V_{n}v^{*}}{v+v^{*}}c_{a}}}\overset{\mathrm{denormalization}}{\overbrace{+\frac{V_{d}v}{v+v^{*}}c_{n}}}$


             $\overset{\mathrm{active}\;\mathrm{motion}}{\overbrace{+\frac{D_{c}}{r^{2}}\frac{\partial^{2}\left(gr^{2}\right)}{\partial r^{2}}}}\overset{\mathrm{advection}}{\overbrace{-\frac{1}{r^{2}}\frac{\partial (I_{s}c_{a}r^{2})}{\partial r}}};$
glucose: 
$\frac{\partial g}{\partial t}=\overset{\mathrm{inflow}}{\overbrace{\left[P_{n}^{g}c_{n}+P_{a}^{g}c_{a}\right]\cdot \left[1-g\right]}}$
(1)
             $\overset{\mathrm{consumption}}{\overbrace{-\left[\left\{\nu_{g}B\right\}n_{p}\Theta_{p}\left(\sigma \right)+Q_{h}^{g}\left\{n_{q}+h+n_{p}\left[1-\Theta_{p}\left(\sigma \right)\right]\right\}\right]\frac{g}{g+g^{*}}}}\overset{\mathrm{diffusion}}{\overbrace{+\frac{D_{g}}{r^{2}}\frac{\partial^{2}\left(gr^{2}\right)}{\partial r^{2}}}};$
free nanoparticles: 
$\frac{\partial u_{f}}{\partial t}=\overset{\mathrm{advective}\;\mathrm{inflow}/\mathrm{outflow}}{\overbrace{\left\{\left[L_{n}\gamma_{n}^{u}\left(\xi \right)c_{n}+L_{a}\gamma_{a}^{u}\left(\xi \right)c_{a}\right]\cdot \left[p_{c}-p\right]\right\}\left[u_{bl}\cdot \Theta \left(p_{c}-p\right)+u_{f}\cdot \Theta \left(p-p_{c}\right)\right]}}$


             $\overset{\mathrm{diffusive}\;\mathrm{inflow}/\mathrm{outflow}}{\overbrace{+\left[P_{n}^{u}\left(\xi \right)c_{n}+P_{a}^{u}\left(\xi \right)c_{a}\right]\cdot \left[u_{bl}-u_{f}\right]}}$

             $\overset{\mathrm{binding}}{\overbrace{-\kappa \left[n_{p}+n_{q}\right]u_{f}}}\overset{\mathrm{lymphatic}\;\mathrm{outflow}}{\overbrace{-L_{l}h\left[p-p_{l}\right]u_{f}}}\overset{\mathrm{diffusion}}{\overbrace{+\frac{D_{u}\left(\xi \right)}{r^{2}}\frac{\partial^{2}\left(gr^{2}\right)}{\partial r^{2}}}}\overset{\mathrm{advection}}{\overbrace{-\frac{1}{r^{2}}\frac{\partial (I_{f}ur^{2})}{\partial r}}};$
bound nanoparticles: 
$\frac{\partial u_{b}}{\partial t}=\overset{\mathrm{binding}}{\overbrace{\kappa \left[n_{p}+n_{q}\right]u_{f}}}\overset{\mathrm{advection}}{\overbrace{-\frac{1}{r^{2}}\frac{\partial (I_{s}ur^{2})}{\partial r}}};$

nanoparticles in blood: 
$\frac{\partial u_{bl}}{\partial t}=\overset{\mathrm{injection}}{\overbrace{\delta (t-t_{u})}}\overset{\mathrm{clearance}}{\overbrace{-C\left(\xi \right)u_{bl}}};$

solid stress: 
$\sigma \equiv \sigma \left(s\right)=k\frac{\left[s-s_{0}\right]{\left[s-s_{s}\right]}^{2}}{{\left[1-s\right]}^{0.1}}\cdot \Theta \left(s-s_{s}\right);$

irradiation: 
$R_{x}=\Gamma_{x}\left(u_{b}\left(r,t\right)\right)\cdot \delta \left(t-t_{R}\right)\cdot n_{x}\left(r,t\right),\;\;x=p,q;$

*where*

$s+f=1,\;\;s=n_{p}+n_{q}+h+m,$



$\Theta_{p}\left(\sigma \right)=\left[1+\tanh\left(\epsilon \left\{\sigma_{p}-\sigma \right\}\right)\right]/2,\;\;\Theta_{tr}\left(g\right)=\left[1+\tanh\left(\epsilon \left\{g-g^{*}\right\}\right)\right]/2,$



$f\left(I_{f}-I_{s}\right)=-K\frac{\partial p}{\partial r},\;\;\frac{\partial p}{\partial r}=-\frac{\partial \sigma }{\partial r},$



$\Gamma_{x}\left(u_{b}\left(r,t\right)\right)=1-exp\left(-k_{y}\left\{\alpha D_{eff}+\beta D_{eff}^{2}\right\}\right),\;\;y=p,q,$



$D_{eff}=\left\{K_{u}u_{b}{\left[\left\{\xi -\psi \right\}/\xi \right]}^{3}\right\}D.$



There are two separate advective motions in the model: $I_{f}=I_{f}\left(r,t\right)$ denotes the absolute advective velocity of the fluid, and $I_{c}=I_{c}\left(r,t\right)$ denotes the advective velocity of the solid phase. When the equations of dynamics of all cells and fluid are added together, the resulting left-hand side becomes zero as a derivative of the constant, and most of the kinetic terms are reduced due to the mass conservation. This leads to implicit Equation (2) for the solid phase velocity, the solution of which depends on boundary conditions, which will be defined further.
(2)$$\begin{array}{c} \frac{\partial }{\partial r}\left(\left[I_{s}-K\frac{\partial p}{\partial r}\right]r^{2}\right)=\left\{\left[L_{n}c_{n}+L_{a}c_{a}\right]\cdot \left[p_{c}-p\right]-L_{l}h\left[p-p_{l}\right]\right\}r^{2}. \end{array}$$

It follows from the first equation that the tumor volume increase caused by the displacement of solid tissue elements ultimately occurs due to the inflow of fluid from capillaries into the tumor mass. The movement of the fluid from the tumor to the normal tissue along the liquid pressure gradient, on the contrary, stimulates tumor shrinkage.

The dynamics of nanoparticles depends on their size as follows.

 Diffusion in tissue:   $D_{u}\left(\xi \right)=D_{u}^{0}/\xi ;$ clearance from blood:  $C_{u}\left(\xi \right)=C_{u}^{0}\xi ;$ fraction of available pore cross-section area:             $\gamma_{y}^{u}\left(\xi \right)=\int_{0}^{X}\left[\pi x^{2}\cdot A\left(\xi ,x\right)\cdot PS_{y}\right]dx/\int_{0}^{X}\left[\pi x^{2}\cdot PS_{y}\right]dx,\;\;y=n,a;$ diffusive permeability:             $P_{y}^{u}\left(\xi \right)=\int_{0}^{X}\left[\pi x^{2}\cdot P\left(\xi ,x\right)\cdot PS_{y}\left(x\right)/N_{y}\right]dx,\;\;y=n,a;$(3) where $A\left(\xi ,x\right)=\Theta \left(x-\xi \right)\cdot {\left(x-\xi \right)}^{2}/x^{2};\;\;P\left(\xi ,x\right)=A\left(\xi ,x\right)\cdot D_{u}\left(\xi \right)\cdot R\left(\xi ,x\right);$       $\;\;\;\;PS_{n}\left(x\right)=x^{3.7}exp\left(-x^{0.9}\right);\;\;PS_{a}\left(x\right)=x^{11}exp\left(-x^{0.85}\right);PS_{b}\left(x\right)=x^{13}exp\left(-x^{0.8}\right);$       $\;\;\;\;R\left(\xi ,x\right)=1-2.1\left(\xi /x\right)+2.09{\left(\xi /x\right)}^{3}-0.95{\left(\xi /x\right)}^{5};$       $\;\;\;\;N_{y}=\int_{0}^{X}\left[\pi x^{2}\cdot P\left(\xi_{g},x\right)\cdot PS_{y}\left(x\right)\right]dx/P_{y}^{g}.$

### 4.2. Parameters

The model contains several dozen parameters, which, if possible, were estimated from the the results of experiments of a different nature or were selected in order to reflect known features of tumor growth. A basic set of parameters is provided in Table 1, where the following normalization parameters are used to derive their model values: $\hat{t}=1$ h for time; $\hat{r}=10^{-2}$ cm for length (except for nanoparticle and pore sizes); $\hat{n}=3\times 10^{8}$ cells/mL for maximum cell density; $\hat{v}=10^{-11}$ mol/mL for VEGF concentration; $\hat{c}=100$ cm${}^{2}$/cm${}^{3}$ for capillary surface area density (based on its average value for human muscle [5], and the rate of glucose consumption by normal cells also corresponds to this tissue); $\hat{g}=1$ mg/mL for glucose concentration; $\hat{D}=1$ Gy for irradiation dose; $\hat{\xi }=1$ nm for nanoparticle and pore size. The normalization factor for nanoparticle concentration is not used explicitly because it is considered to be formally included as a linear multiplier in the factor of dose amplification by the active substance $K_{u}$.

The tumor cell proliferation rate and glucose consumption rates were estimated from the experimental data of the work [26], from which the maximum density of tumor cells $\hat{n}$ was also taken. The rates of these processes were proportionally reduced in order to be more consistent with a less malignant type of tumor, which is justified by the fact that tumor cell invasion is not considered in the model; hence, the malignancy of the tumor in question is not high. The value of critical stress for cell proliferation $\sigma_{cr}$ was estimated by extrapolating data from the work [22], in which a series of experiments were performed on the growth of tumor spheroids subjected to varying external mechanical pressures. The solid stress coefficient was roughly estimated so that the maximum adhesion stress was comparable to the critical stress in absolute values. The smoothing parameter of the Heaviside function was chosen to be large enough to approximate the corresponding functions as step functions, but still to avoid the numerical artifacts. The death rate of damaged cells was equated with the maximum rate of cell proliferation, based on the fact that many cells damaged by radiation die when attempting the next mitosis [4].

The basic value of tissue hydraulic conductivity is taken from the middle part of the range of experimental measurements performed on tumors grown in mice in the work [28] (when considering it on a logarithmic scale). The hydraulic conductivity of abnormal capillaries refers to that of normal capillaries as the ratio of the fractions of their surface areas occupied by pores, which is estimated from Equation (3).

The method of capillary network description via spatially distributed variables imposes certain difficulties on the estimation of parameter values for capillary dynamics, which, however, in practice, strongly depend on a huge number of factors and will differ for different tumor cell lines, host tissues, host organism health states, and will differ from organism to organism. Therefore, we limit ourselves to rough estimates so that the model behavior of the capillary network adequately approximates the general features of the structure and dynamics of the functional capillary network in a tumor during its growth. Capillary density in various mouse tumor models increases three- to six-fold over several days [32], which determines the order of the maximum angiogenesis rate and maximum capillary surface area density. The high-resolution imaging in [6] presents a reconstruction of the microcirculatory bed of a 280 mm${}^{3}$ tumor, which shows that there are very few functioning capillaries within its core. Along with the observation that the capillaries within the tumor degrade within a few days, this allows for the estimation of the rate of capillary degradation and the rate of their active movement [32]. The rate of denormalization is estimated so that the capillaries are almost completely abnormal within the tumor, and the chosen basic rate of normalization ensures that they return to a normal state after a few days, with both observations being based on the work [33]. The Michaelis constant for angiogenesis and denormalization rates is the technical parameter needed to smoothly model the onset and termination of these processes. When VEGF levels are equal to this constant, the rates of these processes are semimaximal. The maximum pore radius was chosen to be large enough so that, for the selected pore spectra, its further increase does not lead to noticeable changes in the values of capillary permeability for nanoparticles with a radius up to 50 nm.

The radiosensitivity of tumor cells was chosen so that in the absence of nanoparticles, when irradiated with a typical dose for clinical protocols, D=2 Gy, approximately half of the cells are damaged. The ratio of tumor cell radiosensitivity parameters α/β is typical for many tumor cell lines [4]. The factor of dose enhancement by the active substance $K_{u}$ was chosen so that the introduction of particles over a wide range of their sizes would, indeed, result in a noticeable dose-enhancing effect in the model. The width of the polymer coating is estimated to be of few nanometers, which is typical according to the order of magnitude of the hydrodynamic radius increase as a result of particle pegylation [40] as well as of the typical size of specific antibodies. The coefficient of binding of nanoparticles to the tumor cells was estimated so that in the presence of tumor cells at their normal density, approximately half of the particles would bind to them in one and a half hours. The parameter for the diffusion coefficient of nanoparticles is based on the characteristic values for the diffusion coefficients of large molecules in the tissue, similar to the case in our work [18].

### 4.3. Numerical Solving

During the numerical simulation of the system of Equations (1)–(3), the equation for intercellular fluid was not explicitly considered due to the equality f=1−s. For the other variables, the method of splitting into physical processes was used, that is, kinetic equations, diffusion equations, and advection equations were solved sequentially at each time step. The kinetic equations were solved by the simple explicit Euler method, which is justified by the relative smallness of the used time steps that are required for the solution of advective equations. For diffusion equations, the implicit Crank–Nicholson scheme was used. These classical methods are described, e.g., in the book [41]. The advective equations were solved using the conservative flux-corrected transport algorithm with an implicit antidiffusion stage, proposed in the work [42]. It is important to note that this method by itself, however, introduces a small amount of uncorrectable diffusion, which leads to the artificial invasion of the tumor into the normal tissue. Another problem of the direct implementation of this algorithm is the inability to correctly model the movement of the free normal tissue boundary on a uniform spatial grid. To overcome these problems, we introduced two additional floating points on the computational grid, marking the position of the tumor–normal tissue interface and the position of the normal tissue boundary. The coordinates of these points were calculated using the conservation of the total cell volume when solving advection equations at each time step.

The following initial conditions were used, representing a spherical section of normal tissue of initial radius $R_{0}^{N}=4$ mm with a small spherical colony of tumor cells of radius $R_{0}^{T}=0.2$ mm located in its center, where r=0:(4)$$\begin{array}{cccc} \left\{\begin{array}{c} n_{p}=s_{st},\\ h=0,\\ g=1,\\ c_{n}=0 \end{array}\right.\; & for\;r\leq R_{0}^{T}; & \left\{\begin{array}{c} n_{p}=0,\\ h=s_{st},\\ g=1,\\ c_{n}=1 \end{array}\right.\; & for\;R_{0}^{T}<r\leq R_{0}^{N}. \end{array}$$

Here, $s_{st}$ is the steady-state value for the fraction of cells differing from $s_{0}$ for the used parameter values by less than a thousandth of a percent, which corresponds to a small stretching of the network of interconnected cells due to the pressure of fluid flowing into the tissue from the capillary system under the blood pressure, but rapidly escaping into the lymphatic system. The values of the other variables were initially equal to zero. Initial conditions are illustrated in Figure 4. Zero-flow boundary conditions were set for all variables on the left boundary; their values on the right boundary were constant and corresponded to the initial conditions for normal tissue. At the left boundary, advective flow velocities were assumed to be zero; at the right boundary, a free boundary condition was used for them. This resulted in the following equation for the velocity of the solid phase of the tissue:$$\begin{array}{c} I_{s}=K\frac{\partial p}{\partial r}+\frac{1}{r^{2}}\int_{0}^{r}\left\{\left[L_{n}c_{n}+L_{a}c_{a}\right]\cdot \left[p_{c}-p\right]-L_{l}h\left[p-p_{l}\right]\right\}z^{2}dz. \end{array}$$

## Figures and Tables

**Figure 1 ijms-24-11806-f001:**
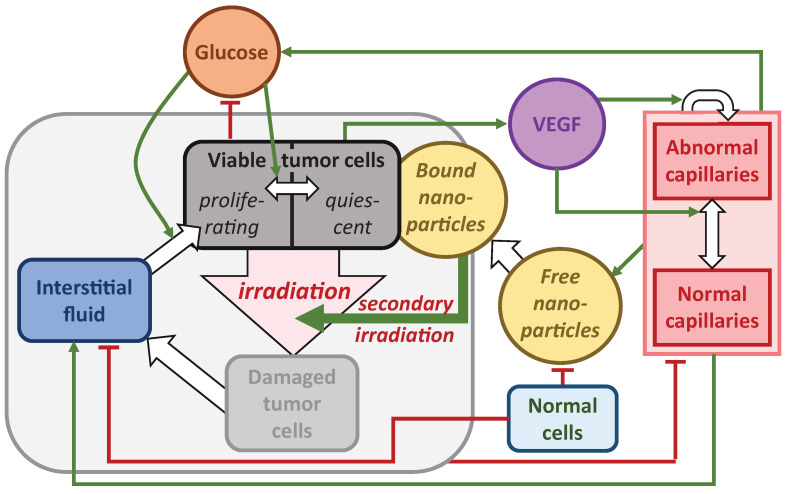
Scheme of the main interactions of the model governed by Equation (1). Green arrows denote stimulating interactions, red lines show inhibiting interactions, white arrows correspond to transitions of variables.

**Figure 2 ijms-24-11806-f002:**
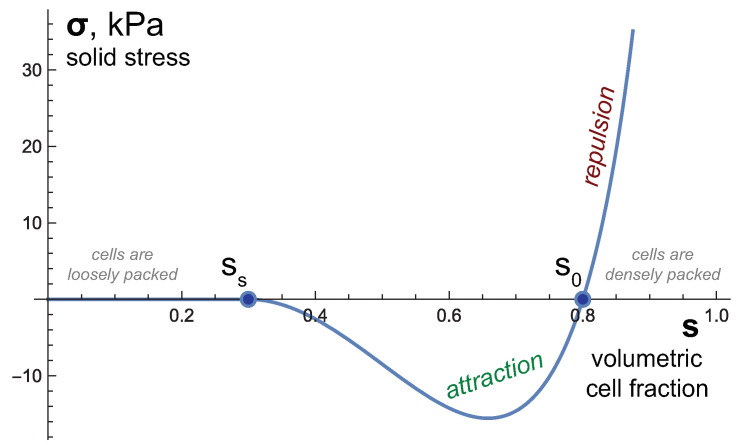
The dependence of solid stress σ on the cell fraction *s* used in the model, governed by Equation (1). The parameter values are based on their basic set provided in Table 1.

**Figure 3 ijms-24-11806-f003:**
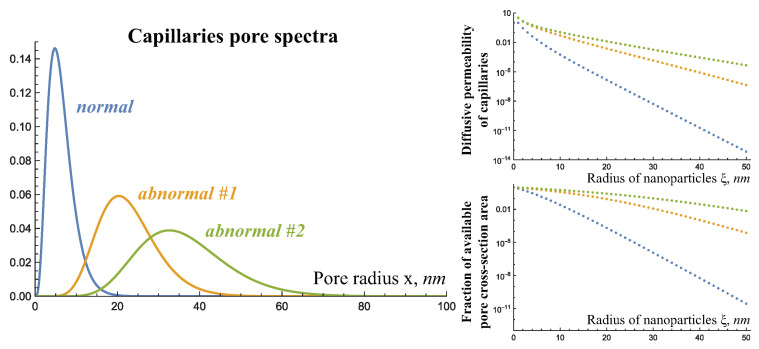
**Left**: normalized pore spectra of capillaries $PS_{n}\left(x\right)$, $PS_{a}\left(x\right)$, and $PS_{b}\left(x\right)$ used in this work and governed by Equation (3). **Right**: resulting values of the diffusive permeability $P_{y}^{u}\left(\xi \right)$ and the fraction of the pore cross-sectional area $\gamma_{y}^{u}\left(\xi \right)$ available for nanoparticles in the walls of capillaries of different types.

**Figure 4 ijms-24-11806-f004:**
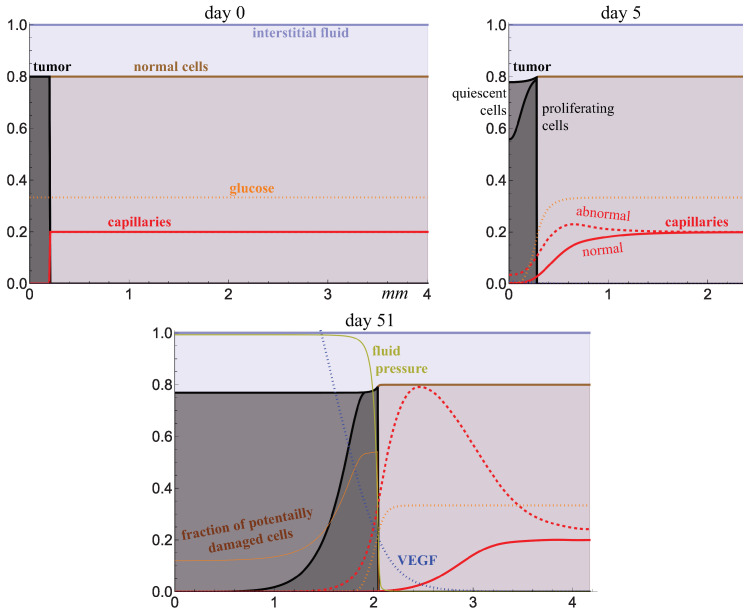
Distributions of model variables obtained from numerical simulations of free tumor growth, Equation (1). Values of the variables for glucose and capillaries are renormalized for better visualization.

**Figure 5 ijms-24-11806-f005:**
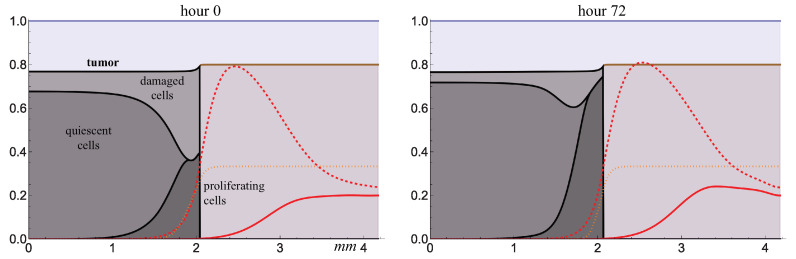
Distributions of model variables obtained by numerical simulations after irradiation of tumor on day 51 of its growth. Styles of the lines correspond to that in Figure 4.

**Figure 6 ijms-24-11806-f006:**
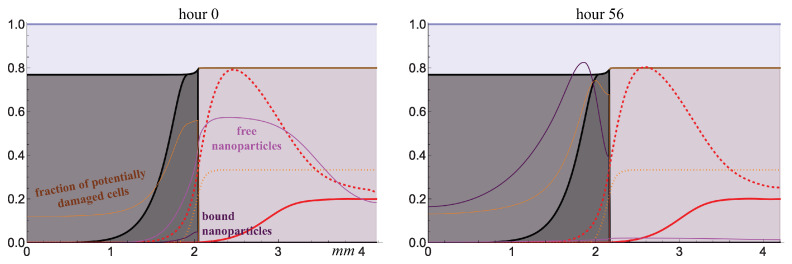
Distributions of model variables obtained by numerical simulations after the injection of nanoparticles into the blood (ξ=20) on day 51 of tumor growth. Styles of the lines correspond to that in Figure 4.

**Figure 7 ijms-24-11806-f007:**
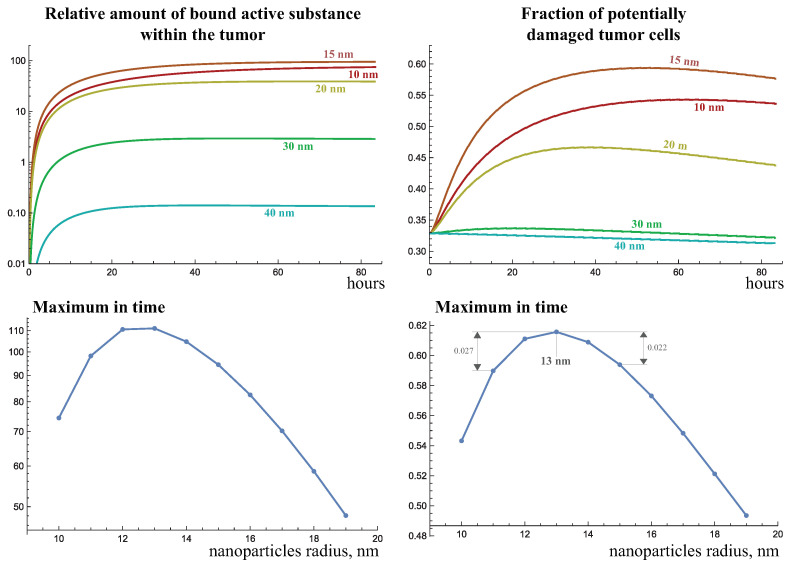
Dynamics of the bound active substance and the fraction of potentially damaged tumor cells when nanoparticles of varying size are injected at day 51 of tumor growth under the abnormal capillary pore spectrum $PS_{a}\left(x\right)$, defined in Equation (3).

**Figure 8 ijms-24-11806-f008:**
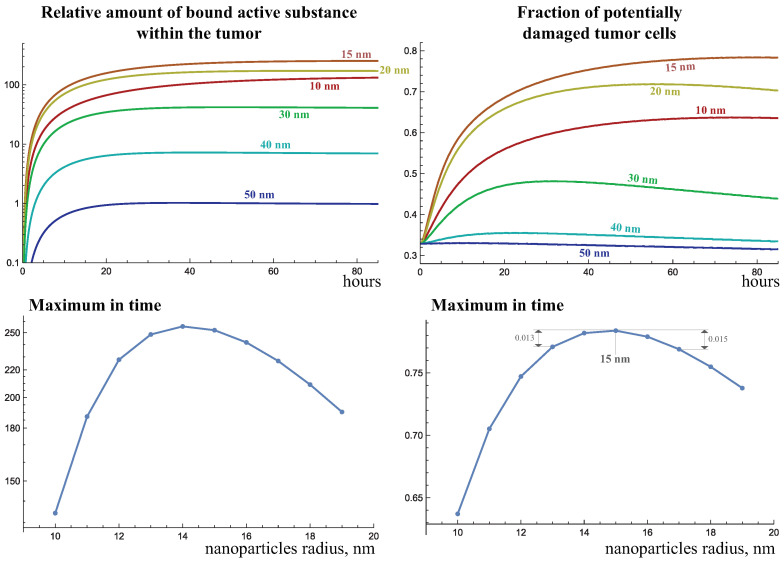
Dynamics of the bound active substance and the fraction of potentially damaged tumor cells when nanoparticles of varying size are injected at day 51 of tumor growth under the abnormal capillary pore spectrum $PS_{b}\left(x\right)$, defined in Equation (3).

**Figure 9 ijms-24-11806-f009:**
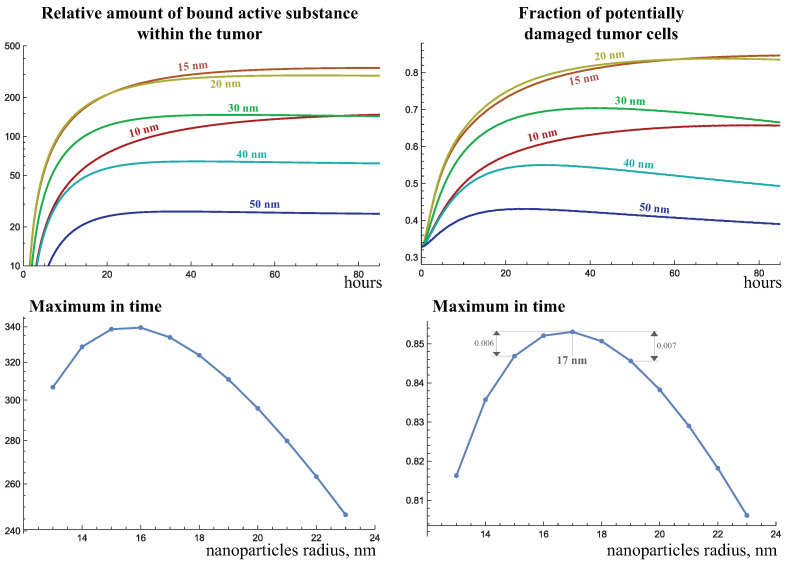
Dynamics of the bound active agent and the fraction of potentially damaged tumor cells when injecting nanoparticles of varying size at day 51 of tumor growth with a uniform pore spectrum of abnormal capillaries.

**Table 1 ijms-24-11806-t001:** Model parameters.

Parameter	Description	Value	Based on
**Cells:**			
*B*	maximum rate of cell proliferation	0.01	[26]
$\sigma_{p}$	critical stress for cell proliferation	15	[22]
ϵ	smoothing parameter of Heaviside function	500	see text
*M*	the rate of death of damaged cells	0.01	see text
**Stress:**			
*k*	solid stress coefficient	500	see text
$s_{s}$	minimum fraction of interacting cells	0.3	[27]
$s_{0}$	initial fraction of cells	0.8	[27]
**Interstitial fluid:**			
$L_{n}$	hydraulic conductivity of normal capillaries	0.1	[13]
$L_{a}$	hydraulic conductivity of abnormal capillaries	0.22	see text
$p_{c}$	fluid pressure in capillaries	4	[13]
$L_{l}$	hydraulic conductivity of lymphatic capillaries	1300	[13]
$p_{l}$	lymph pressure	0	[13]
*K*	tissue hydraulic conductivity	0.1	[28]
**VEGF:**			
$S_{v}$	secretion rate	1	[29]
ω	internalization rate	1	[30]
$M_{v}$	degradation rate	0.01	[31]
$D_{v}$	diffusion coefficient	21	[31]
**Capillaries:**			
*R*	maximum rate of angiogenesis	0.008	[32]
$c_{max}$	maximum surface area density	5	[32]
$M_{c}$	characteristic degradation rate	0.03	[6,32]
$k_{M}$	coefficient of degradation in the tumor core	2	[6,32]
$V_{n}$	normalization rate	0.1	[33]
$V_{d}$	denormalization rate	0.1	[33]
$v^{*}$	Michaelis constant for VEGF action	0.001	see text
$D_{c}$	coefficient of active movement	0.03	[6,32]
*X*	maximum pore radius	100	see text
**Glucose:**			
$g^{*}$	Michaelis constant for consumption	0.01	[34]
$P_{n}^{g}$	permeability of normal capillaries	4	[35]
$P_{a}^{g}$	permeability of abnormal capillaries	10	[18]
$\nu_{g}$	parameter of consumption by proliferating cells	1200	[26]
$Q_{h}^{g}$	rate of consumption by normal tissue	0.5	[36]
$D_{g}$	diffusion coefficient	100	[37]
$\xi_{g}$	hydrodynamic radius	0.36	[38]
**Irradiation:**			
α	linear parameter of cell radiosensitivity	0.1	see text
β	quadratic parameter of cell radiosensitivity	0.01	[4]
$k_{p},\;k_{q}$	coefficients for proliferating and quiescent cells	1, 0.2	[39]
*D*	irradiation dose	2	see text
$K_{u}$	factor of dose enhancement by radiosensitizer	100	see text
**Nanoparticles:**			
κ	coefficient of binding with tumor cells	0.5	see text
ψ	width of polymer coating	7	see text
$D_{u}^{0}$	parameter for diffusion coefficient	65	[18]
$C_{u}^{0}$	parameter for clearance rate	0.003	see text

## Data Availability

The data presented in this study are available on request from the corresponding author.

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
