# Peer review of "Optimization of Size of Nanosensitizers for Antitumor Radiotherapy Using Mathematical Modeling"

_ijms, 2023, doi:10.3390/ijms241411806_

Round 1

Reviewer 1 Report

The work describes treatment of tumors with nanoparticles capable of emitting secondary radiation under radiation therapy. Cancer and its biomechanics are not my field, hence I cannot pass any judgement on the validity or importance. Still, as a biomodeler, I would like to make suggestions regarding the general structure:
1. The authors propose a rather complex model of tumor mechanics with radial symmetry, but without any diagram.  They must include several diagrams showing compartments, flow, interstitials,  to let the reader to understand these equations in geometric terms. 2. Do such simple curves in the end require such a complex model? Can some factors and terms be dropped? "The less the better" is usually the principle for modeling. 3. Terminology: a. What does "convection" mean? In physics, it is movement of liquid in the presence of a temperature gradient and a gravitational field. And here? b. Are nanoparticles covered, by tumor-specific antigens, which does not make sense, or tumor-specific homing molecules?

Optimization of nanosensitizers size

should read

Optimization of nanosensitizer size

or even better

Optimization of the size of nanosensitizers

Author Response

We would like to thank the reviewer for the provided fruitful remarks. Below are our point-to-point replies:

1. Yes, presenting a diagram is valuable for the readers' comprehension. We have now included a block-scheme of the main model interactions as Fig. 1.

2. Surely, "as simple as possible but not too simple" is a general principle for successful models able to yield reliable predictions. However, this principle rather applies to the search for qualitative concepts, that provide general insights into the models behavior under broad range of conditions, and that reveal the optimized methods of controling them (a prominent example in mathematical oncology is adaptive therapy which suggests adjustment of drug infusion rate on the basis of on ongoing evolution of resistant populations within the tumor). In our current work we however deal with the problem of another type and and we aim to provide a quantitative answer. Therefore, it is reasonable to account for all the known physiological processes, that can eventually influence the distribution of nanoparticles within the tumor under variation of their size, as well as to account for spatial distribution of model variables. Note, for example, that the nonmonotonic nature of the dynamics of fraction of potentially damaged tumor cells is guided namely by the relative motion of cells, bound nanoparticles and glucose concentration fields.

We have now clarified this issue in the fourth paragraph of the introduction.

3.a. While it could be argued that the term ``convection'' has a broad meaning, it is indeed generally associated with the fluid motion due to temperature gradients. In order to avoid ambiguity, we have replaced this term with ``advection'', which is quite a common term for the use in the field of mathematical oncology in the similar tasks.

3.b. That is an unfortunate mistake stemmed from the translation of original text, thank you for pointing it out. The nanoparticles are covered by tumor-spesific antibodies, we have now corrected it.

Reviewer 2 Report

See my attached referee report.

Author Response

We would like to thank the reviewer for the provided fruitful remarks. Below are our point-to-point replies:

1. We have corrected the keywords.
2,3,6. We have corrected the reference style.
4. We have now clarified that it is an implicit eqaution for solid phase velocity, which solution depends on boundary conditions, that will be defined further. Therefore yet it has non-explicit form.
5. We have included additional explanations in the Figure 1, aimed to clarify that the solid stress function accounts for the following phenomena: when the fraction of cells is normal, cell interactions result in zero solid stress; as the cells get closer together there is a repulsive interaction; as the distance between the cells increases, an attractive interaction emerges which at first intensifies and then gradually weakens due to successive ruptures of individual intercellular contacts.
7. We have mentioned the papers, related to the optimization tasks, in the introduction, in order to highlight the difference between the optimization tasks in life sciences versus exact sciences and computer science, as follows: "Ideally, the corresponding optimization tasks have to imply the enormous complexity and dramatical variability of cancers, their constant evolution and impossibility of performing reliable estimations of all the related parameters during an ongoing treatment. That drastically distinguishes optimization tasks in mathematical oncology from the optimization tasks in exact sciences and computer science." Since the indicated papers are not related to the field of mathematical oncology, comparing our results with the ones presented therein does not seem plausible. 

Round 2

Reviewer 1 Report

Thanks for including a diagram.

I completely disagree that a quantitative study must include all known processes. This is a popular mistake of a novice. In fact, such an approach  lowers the accuracy and makes the system over-defined. Its quantitative predictive value is zero.

The complexity of the model should be defined by the question asked in the study, and robustness to simplifications must be tested carefully.

The bright side is that the authors did all the work of collecting all these processes and parameters from the literature, and some had to do it.